# Study on the Stability of Cut Slopes Caused by Rural Housing Construction in Red Bed Areas: A Case Study of Wanyuan City, China

**Hailin He [1,2], Xiujun Dong [1,\*], Simin Du [2], Hua Guo [2], Yue Yan [2] and Guohui Chen [2]**

[1]   State Key Laboratory of Geo-Hazard Prevention and Geo-Environmental Protection,
     Chengdu University of Technology, Chengdu 610059, China; 2022010118@stu.cdut.edu.cn
[2]   The 1st Geological Brigade of Sichuan, Chengdu 610072, China
\*   Correspondence: dongxiujun@cdut.cn; Tel.: +86-028-87732191

**Abstract:** The red beds that are widely distributed in hilly areas in eastern Sichuan Province are inevitable rock and soil masses in engineering construction. In order to obtain a larger building area, engineering slope cutting is quite common in red bed hilly areas. Engineering slope cutting destroys the inherent stability of a slope and causes geological disasters. In order to practice the concept of sustainable development and explore ways to develop and utilize land resources reasonably and reduce the possibility of geological disasters caused by engineering slope cutting, this paper took the slope cutting sites caused by rural housing construction in the red bed area of Wanyuan City as research objects. The internal and external factors affecting the stability of the cut slopes were summarized through a field investigation, and two typical slopes were selected for analysis. Sampling and indoor tests were conducted to obtain the geotechnical parameters. Geo Studio software (2018 R2) was employed to establish numerical models, simulate the stress and strain distributions, and compute stability coefficients under different slope cutting conditions at the same time. Based on field investigations and numerical simulations, the three main failure modes of shallow landslides caused by slope cutting were summarized, and the evolution process of slope landform accelerated by slope cutting activities was deduced. In an engineering application, the functional relationship between the cutting height and the stability coefficient was fitted. It was found that the critical cut height values of soil slope were 6.3 m, 6.2 m, 5.2 m, and 2.6 m at slope of 10°, 20°, 30° and 40°, respectively; the critical cut height values of rock–soil mixed slope were 9.3 m, 6.5 m, 5.9 m, and 2.2 m at slope of 10°, 20°, 30° and 40°, respectively. The research findings can be used to prevent and manage the hazards caused by slope cutting in this study area.

**Keywords:** slope cutting; numerical simulation; stability coefficient; red bed; critical value

## 1. Introduction

Red beds are mainly red clastic sedimentary stratums which can not be avoided in engineering construction in eastern Sichuan Province. With the continuous development of urban and rural construction in red bed areas, slope cutting, which can increase the building area in some rural mountainous regions, is becoming more and more common. The high and steep cut slopes formed by artificial excavation are prone to geological disasters, such as landslides and collapses, that always lead to deaths and property damage [1–3]. Many scholars around the world have studied the stability of natural and excavated slopes and landslides.

The stability of a slope is influenced and controlled by many factors [4,5]. It has been found that the internal friction angle φ has the greatest influence on slope stability, followed by the horizontal seismic coefficient Kh, cohesion c, and saturation permeability ks [6]. Shepheard C J and others take the standpoint that the determination of slope safety usually needs to consider other factors, such as the slope geometry, soil mechanical

characteristics, and groundwater conditions [7]. Many hidden factors of landslides have been discovered, including internal factors, such as the geological background, slope structure type, material combination, and landform, and external factors, such as rainfall, earthquakes, and snow melting [8–10]. Strouth and McDougall applied the risk prediction value to investigate the prevention and control of landslides [11].

A large number of scholars have focused on the stability and safety of cut slopes. Panthee studied the relationship between the slope stability factor and the height and gradient of engineered cutting slopes [12]. It is particularly important to limit the height and gradient of a cut slope according to its structure [13]. The instability volume of a loess slope is affected by the slope gradient and height, and the larger the slope and height are, the larger the instability volume is [14]. The stability of a slope is controlled by key blocks, and the slope toe plays a vital role in maintaining slope stability [15]. Zhang and Wang found that the critical conditions for the propagation of catastrophic shear zones on engineering slopes are related to the gravitational shear stress ratio of the slope, which is mainly controlled by the slope toe [16]. The research shows that the stability of soil agglomerates is very important to the soil erosion resistance of an engineered graben slope in geologically hazardous areas, thus indicating that the rock strength and soil erosion resistance are important factors in determining the slope stability [17,18]. According to Awang and others, a rock is exposed after slope cutting, and its strength decreases over time due to weathering [19]. According to a study of the evolution of the disaster-causing slope cutting of red layer slopes, the horizontal displacement is larger than the vertical displacement, the deformation gradually increases with the gradation of the cut slope, and the deformation area of the slope is primarily concentrated near the cut slope surface [20].

Quantitative analysis has always been the focus of slope stability research, and it is extensively employed to determine the dangers of landslides [11,21,22]. AYBE RK KAYA combined kinematics, limit equilibrium, and numerical stability analyses to study the failure mechanism of houses on nearby slopes after the excavation of basalt and tuff tunnels; they confirmed that the slope instability is related to the strength reduction in rock mass and joints after rock mass disturbance [23]. TieSheng Yuan et al. utilized Geo Studio to establish a deformation model to validate the conformity of a numerical method and investigate a cut slope [24].

In China, with the in-depth implementation of the concept of sustainable development and safe development in recent years, all levels of governments have taken measures to strengthen the safety management of engineering slope cutting; so, methods that can effectively reduce the possibility of disasters caused by slope cutting are urgently needed. In order to explore a practical reference scheme for slope cutting, the red beds in Wanyuan City were taken as the research area. The main factors affecting the stability of the slope cutting site caused by rural housing construction were investigated and analyzed. Two-dimensional numerical models of the typical cut slopes were established, and the stress–strain redistribution and stability coefficient under different slope cutting conditions were simulated and computed. The functional relationship between the slope stability coefficient and slope cut height was fitted, and the critical and recommended slope cutting heights were determined.

## 2. Study Area

### 2.1. Overview of the Study Area

Wanyuan City is located 150 km northeast of Dazhou City, Sichuan Province, where Sichuan Province meets Shanxi Province (Figure 1a). The overall topography of Wanyuan City is characterized by high elevations in the northeast and low elevations in the southwest. The northeast region is mainly mountainous and composed of canyons. Most areas in the middle, southwest, and northwest are middle-elevation and low-elevation mountain platform landforms. The southern part is mainly platform canyons. Red beds are distributed in the central and western areas in Wanyuan City (Figure 1c). The flat areas in Wanyuan City are basically occupied by engineering construction or basic farmland

that cannot be occupied; the flat areas used to build rural housing on are smaller, so slope cutting is quite common.

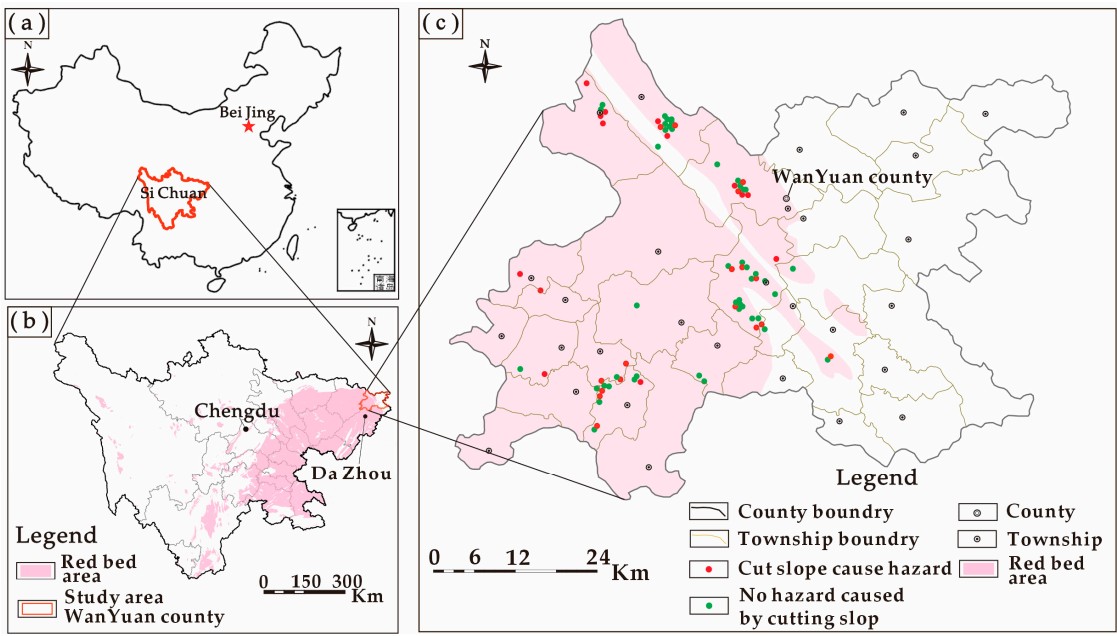

**Figure 1.** Overall of slope cutting sites caused by rural housing construction in study area. (**a**) the location of Sichuan Province; (**b**) the location of study area and distribution of red beds in Sichuan Province; (**c**) the distribution of 76 slope cutting sites caused by rural housing construction in red beds of Wanyuan county.

### 2.2. Characteristics of Cut Slopes

From April to October, 2022, the research team investigated 76 red bed slope cutting sites in the study area. It can be found from Figure 1c that the distribution of slope cutting sites is quite uneven on the horizontal plane. The sites are densely distributed in some villages and towns with limited flat land resources, while fewer or no sites exist in some villages and towns with relatively more flat land resources. From the perspective of topography, influenced by factors such as capital, technology, and construction machinery, the slope cutting sites are mostly located at the foot of slopes where the Quaternary cover is relatively thick.

### 2.3. Factors on the Cut Slope Stability

In order to further analyze the influence of internal and external affecting factors on the stability of cut slope caused by rural housing construction, the research team conducted a detailed investigation on 31 disaster-causing slope cutting sites in the study area (red points in Figure 1c). The statistics show that the internal factors affecting the stability of cutting slopes mainly include the slope type, lithologic composition, and natural slope gradient, and the external factors include the cut slope height, gradient, and protection engineering.

Figure 2 shows the investigation results of 31 disaster-causing slope cutting sites generated by rural housing construction in the research area. The cutting slopes are mainly composed of red mudstone, shale, and sandstone, accounting for 75%. The slope types include soil slopes (accounting for 52%), rock–soil mixed slopes (accounting for 26%), and rock slopes (accounting for 22%). The gradients of these natural slopes are 15–25° (accounting for 39%), 25–35° (accounting for 28%), and 5–15° (accounting for 17%), respectively. The cut slope heights are mainly 8–10 m (accounting for 36%) and 6–8 m (accounting for 29%). In order to obtain larger building areas, most of the cut slopes are excavated vertically, and the cut slope gradients exceeding 85° account for 94%. Twenty-nine (accounting for 94%) cut slopes have not been subjected to engineering, or the treatment measures have been

insufficient, while 55% of the cut slopes are in an unstable state. Landslides are the main type of hazard caused by slope cutting (accounting for 81%), which mainly occurs in soil and rock–soil mixed slopes.

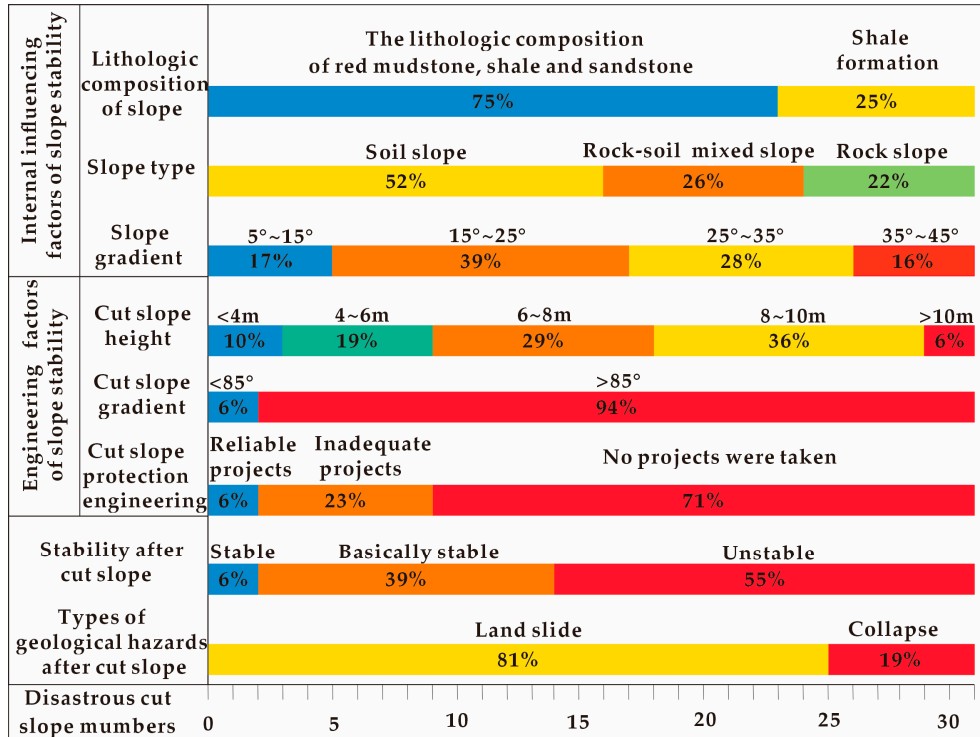

**Figure 2.** Statistics on stability of disastrous cut slopes.

## 2.4. Typical Slope Cutting Sites

Based on a field investigation and the analysis of the internal and external factors of the disaster-causing cut slopes generated by rural housing construction, this paper studied the stability of typical soil and rock–soil mixed slopes, respectively. The soil and rock–soil mixed slopes are defined according to the thickness of the overburden and the influence of slope cutting activities on the overlying soil and underlying bedrock. A soil slope has a thickness of 5–15 m, and slope cutting activities only effect the overlying soil layer. A rock–soil mixed slope has an overlying soil layer thickness of 1–5 m, and slope cutting cuts through the soil layer and destroys the bedrock.

Figure 3a shows a typical soil slope cutting site, which is located behind Chen Qigang's house in Wenjiaping Village in Baiguo Town. This gentle slope is on a low-elevation mountain, and cutting was performed at the foot of the slope. The slope gradient is about 20°. The cut height is about 8.5 m, the inclination is 190°, and the cut gradient is 83°. Slope cutting revealed that the surface covering the stratum is Quaternary eluvial silty clay, which is brownish-yellow and contains plant roots. Soil slopes are widely distributed in the study area and have the following common characteristics: (1) the Quaternary overburden at the foot of soil slope is often thick; (2) slope cutting is convenient, but the slope is unstable; (3) usually, there is no unified sliding surface during a failure, and the sliding scale is relatively small.

Figure 3b shows a typical rock–soil mixed slope, which is located in Maoping Village in Jingxi Town. This gentle slope is also on a low-elevation mountain, and cutting was performed at the foot of the slope too. The natural slope gradient is less than 20°. The cut height is about 13 m, the inclination is 205°, and the cut gradient is 65° at the top and 85° at the bottom. Slope cutting revealed that the surface covering the stratum is Quaternary eluvial silty clay, which is yellow-brown and 2.5 m thick, and the underlying bedrock is layered sandstone; the rock stratum is nearly horizontal. Compared with the soil slope

damaged by slope cutting, the landslide volume of the rock–soil mixed slope is larger, so the damage would be relatively worse. In August, 2020, a landslide with a volume of about 200 m³ occurred at this slope cutting site, which damaged the house at the toe of this slope.

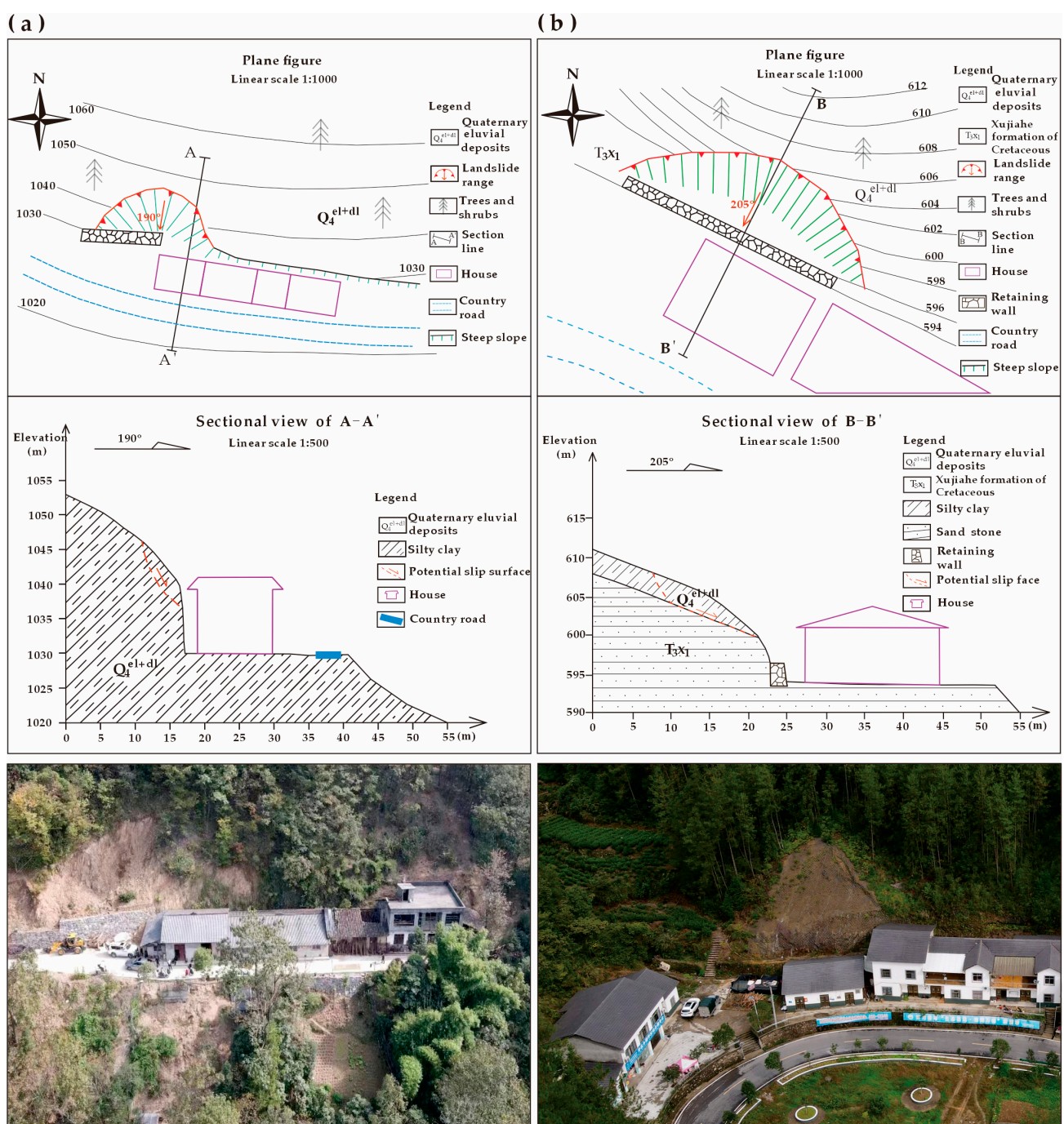

**Figure 3.** Typical slop cutting sites. (**a**) Soil slope; (**b**) rock–soil mixed slope.

## 3. Materials and Methods

The constitutive models and geotechnical parameters were determined based on the collection and analysis of geological materials and reports, as well as field investigations and laboratory tests. Geo Studio software were used for numerical model construction and simulation. The flowchart of the methodology is shown in Figure 4. The details of the analysis are as follows.

### 3.1. Establishment of Constitutive Model

An investigation of all 76 slope cutting sites caused by rural housing construction in red beds of Wanyuan city were conducted by the research team. According to the investigation results, 31 of 76 slope cutting sites had failure records, which were called disaster-causing cut slopes, and even damaged houses at the toe of slope. Soil slope and rock–soil mixed slope accounting for 52% and 26%, respectively, are the main types of 31 cut slopes, and were selected as the two main constitutive model type. According to the local people's experience for many years and the actual investigation result, the slope cutting site of rural housing construction always locates at the toe of slope where both the topography and lithologic composition are simple. In this paper, the two-dimensional profiles of 16 soil cut slopes and 8 rock–soil mixed slopes were simplified as constitutive models, the geometric shapes of which are presented in Figure 5.

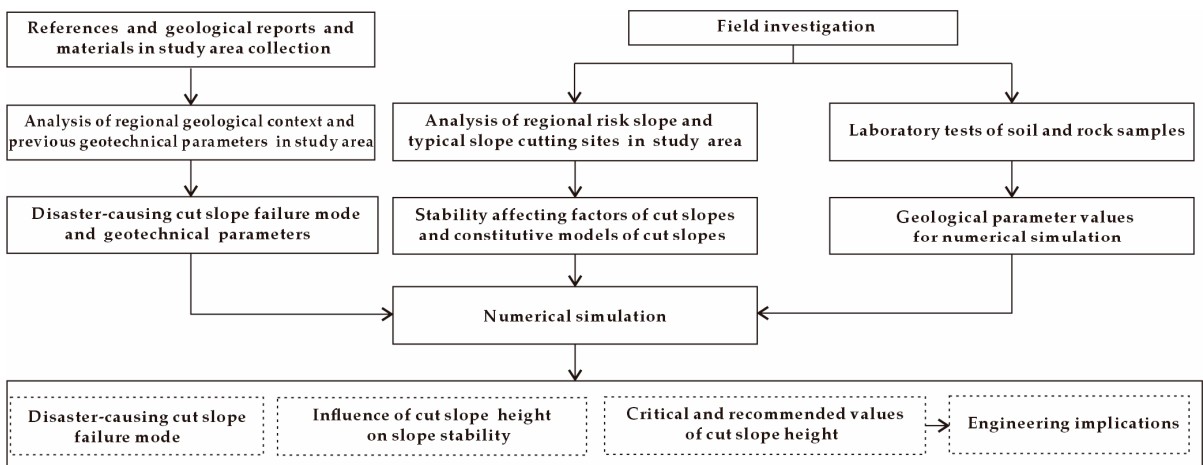

**Figure 4.** Flowchart of the methodology.

### 3.2. Geological Parameter Values

The soil and rock samples were collected in the red beds area of Wanyuan city via field investigation and engineering drilling (Figure 5). The geotechnical parameters of the samples were analyzed, including the density, cohesion, internal fiction angle, Poisson's ratio, and elastic modulus. All test methods and steps followed the "Geotechnical test method standard (GB/T 50123-2019)" [25]. The historical geotechnical parameters of soil and rock were collected from a local government department at the same time. These geotechnical parameters were obtained from different times at different positions in red beds of Wanyuan city. Average values (Table 1) were used to construct numerical models.

The main type of rock is sandstone in the study area (Figure 5e). According to the laboratory analysis of the sandstone drilling samples, the content of sand and gravel (2–0.05 mm) is generally more than 50%, followed by silt and clay. The cements of sandstone are siliceous, calcareous, calcareous argillaceous, and argillaceous, and the carbonate content is about 20–25%. Sandstone has low strength, strong water permeability, and easy weathering after excavation. The main average geotechnical parameter values of moderately weathered sandstone are shown in Table 1.

**Table 1.** Geotechnical parameters used in numerical simulation.

| Materials | Density (KN/m$^3$) | Cohesion (Kpa) | Internal Friction Angle (°) | Poisson's Ratio | Elastic Modulus (Mpa) |
|---|---|---|---|---|---|
| Eluvial silty clay | 18.5 | 18 | 21.5 | 0.3 | 30 |
| Moderately weathered sandstone | 25.7 | 2750 | 39.6 | 0.25 | 7000 |

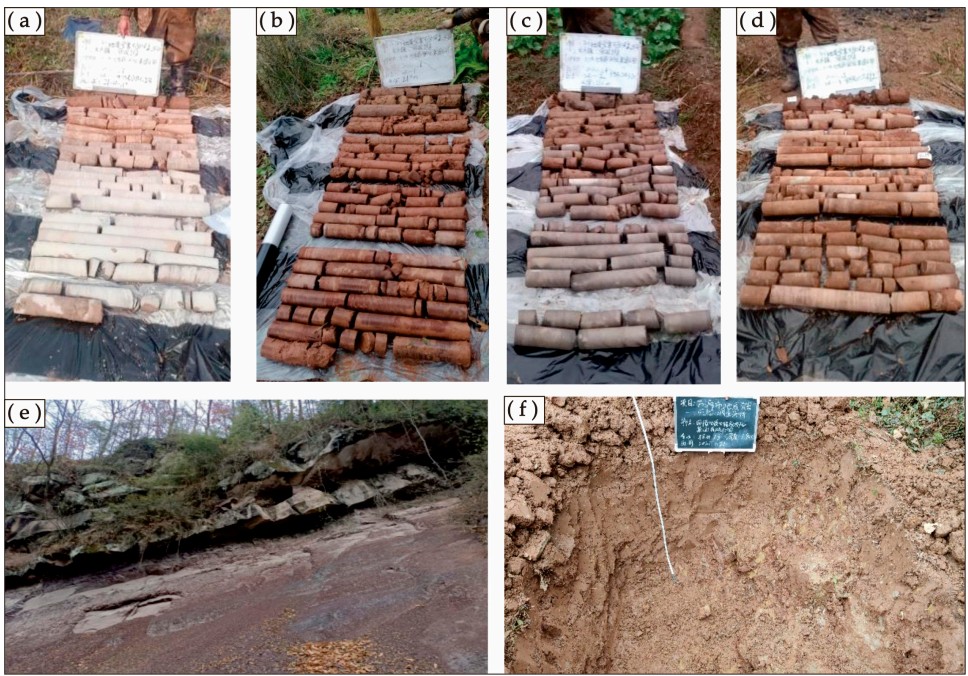

**Figure 5.** Soil and rock samples obtained from investigation, drilling, and prospecting trench. (**a**–**d**) Samples from drilling; (**e**) moderately weathered sandstone; (**f**) prospecting trench.

### 3.3. Numerical Model Construction of Typical Cut Slopes

In order to fully simulate the characteristics of stress and strain of soil and rock–soil mixed slopes during slope cutting, Geo Studio software was employed to construct numerical models with 10°, 20°, 30°, and 40° natural slope gradients of these two typical cut slopes, respectively.

The specific modeling steps are as follows:

1.  Model area setting: To control a single variable, the main profile of the cut slope was selected as the model area, the slope height of soil slope was set to be 35 m, and the slope height of rock–soil mixed slope was set to be 40 m. Slope models with different slope gradients were constructed by changing the horizontal distance from the toe to the top of slope, and the different slope cutting conditions were simulated by changing the excavation heights at the toe of the slope.

2.  Determination of the constitutive model: The soil slope was generalized as a Mohr–Coulomb model composed of homogeneous eluvial silty clay layers. The rock–soil mixed slope was generalized as a Mohr–Coulomb model consisting of a eluvial silty clay layer and moderately weathered sandstone bedrock.

3.  Boundary condition setting: The model had no horizontal displacement in the ordinate direction and no vertical displacement in the abscissa direction.

4.  Regional grid division: To ensure the best precision, accuracy, and calculation speed of the finite element method, the grid spacing was set to 1 m, and quadrilateral and triangular elements were used to connect the two materials in the transition area, so that each area of the model was divided into multiple independent cells.

The two-dimensional numerical models of the soil and rock–soil mixed slopes obtained through the above modeling steps are shown in Figure 6, and this paper only takes the 20 slope gradient as an example to present.

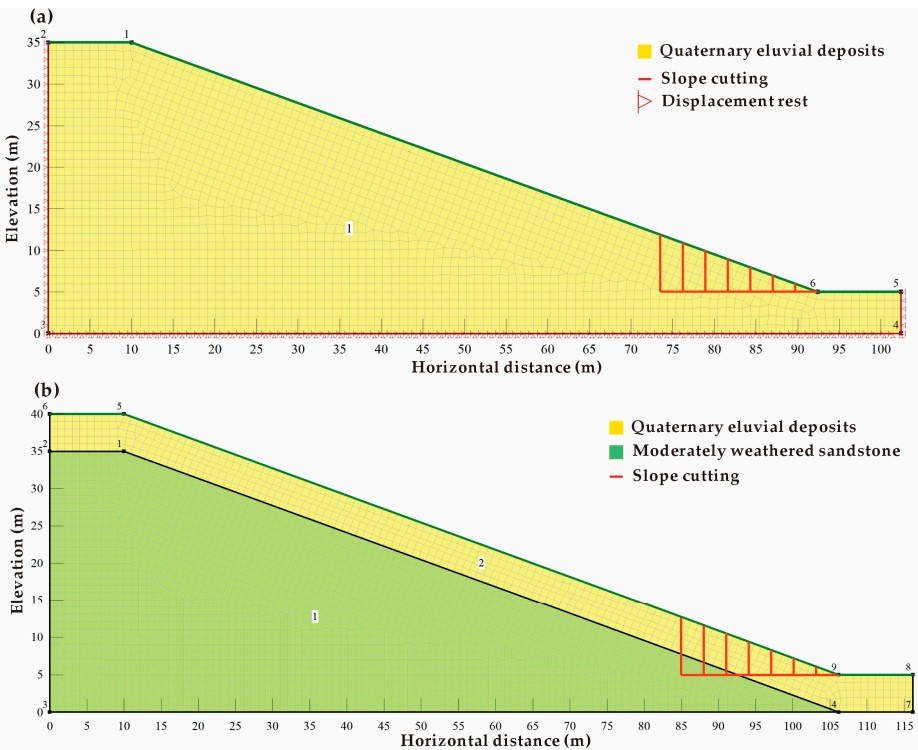

**Figure 6.** The two-dimensional numerical model of 20 natural slope. (**a**) Numerical model of soil slope; (**b**) numerical model of rock–soil mixed slope.

*3.4. Numerical Simulation of Different Slope Cutting Conditions*

In this paper, Slope/W and Sigma/W modules in Geo Studio software were used for the numerical simulation. The combination of these two modules was used for the comprehensive analysis, which mainly included the following two major steps: (1) the finite element method of Sigma/W module was used to analyze the stress and strain redistribution of the slopes under different excavation conditions; (2) the results of stress and strain redistribution were imported into Slope/W module, and the stability coefficient in equilibrium state and the distribution of potential instability zones were obtained using the limit equilibrium method.

The Mohr–Coulomb ideal elastic–plastic model was adopted in the calculation process. Firstly, the slope body in the un-excavated state was simulated and calculated, and then it was excavated from the toe of the slope at the elevation of 5 m on the longitudinal coordinate axis. The cutting height was 0–7 m, and the stress–strain distribution and the stability coefficient of the slope were simulated and calculated at every 1 m increase in the cutting height.

## 4. Results

*4.1. Numerical Simulation Results*

The numerical simulated horizontal stress values and the stability coefficients of the cut slopes are shown in Tables 2 and 3, respectively.

*4.2. Stress–Strain Redistribution Caused by Slope Cutting*

The horizontal stress values at the toes of soil and rock–soil mixed slopes under different slope cutting conditions are shown in Table 2. The horizontal stress at the toes of these two slopes are positively correlated with both the natural slope gradient and the cutting height; as the cutting height and slope gradient increases, the horizontal stress at the toe increases. It was found that the horizontal stress at the toe of the rock–soil mixed slope generally increases faster than that of the soil slope, and the greater the slope gradient is, the higher the cutting height is, and the more obvious this phenomenon is.

**Table 2.** Numerical simulation results of horizontal stress values at the toe of slope under different slope cutting conditions.

| Gradient of Natural Slope/(°) Horizontal Stress Values/(kp) Cut Height/(m) | | 0 | 1 | 2 | 3 | 4 | 5 | 6 | 7 |
|---|---|---|---|---|---|---|---|---|---|
| Soil slope | 10° | 8.4 | 14.2 | 18.4 | 28.8 | 34.5 | 40.1 | 47.1 | 55.4 |
| | 20° | 9.6 | 19.2 | 28.7 | 47.9 | 57.5 | 65.3 | 76.6 | 86.2 |
| | 30° | 38.3 | 51.3 | 65.7 | 79.6 | 92.3 | 105.3 | 112.6 | 137.6 |
| | 40° | 47.9 | 57.5 | 67.0 | 86.2 | 105.3 | 117.4 | 134.1 | 153.2 |
| Rock–soil mixed slope | 10° | 6.7 | 7.5 | 8.6 | 10.1 | 13.2 | 18.0 | 23.2 | 31.5 |
| | 20° | 10.2 | 15.4 | 23.8 | 32.5 | 40.3 | 46.0 | 53.2 | 66.4 |
| | 30° | 20.2 | 38.2 | 61.4 | 82.9 | 101.7 | 123.5 | 142.8 | 155.7 |
| | 40° | 38.2 | 67.9 | 99.7 | 115.2 | 147.8 | 162.5 | 183.2 | 201.5 |

**Table 3.** Numerical simulation results of stability coefficients under different slope cutting conditions.

| Gradient of Natural Slope/(°) Stability Coefficients Cut Height/(m) | | 0 | 1 | 2 | 3 | 4 | 5 | 6 | 7 |
|---|---|---|---|---|---|---|---|---|---|
| Soil slope | 10° | 2.853 | 2.536 | 2.224 | 1.941 | 1.381 | 1.174 | 1.009 | 0.956 |
| | 20° | 1.601 | 1.596 | 1.577 | 1.467 | 1.249 | 1.162 | 1.005 | 0.898 |
| | 30° | 1.128 | 1.124 | 1.097 | 1.087 | 1.055 | 1.023 | 0.993 | 0.835 |
| | 40° | 1.047 | 1.034 | 1.002 | 0.964 | 0.869 | 0.745 | 0.677 | 0.278 |
| Rock–soil mixed slope | 10° | 3.865 | 3.654 | 3.326 | 3.155 | 2.865 | 2.550 | 2.123 | 1.895 |
| | 20° | 2.223 | 2.111 | 1.949 | 1.756 | 1.534 | 1.281 | 1.025 | 0.945 |
| | 30° | 1.238 | 1.219 | 1.193 | 1.159 | 1.117 | 1.066 | 0.977 | 0.932 |
| | 40° | 1.048 | 1.037 | 1.008 | 0.974 | 0.932 | 0.884 | 0.901 | 0.887 |

In order to further analyze the influence of slope cutting on the stability of a soil slope, this paper took 20 soil slopes as an example to compare and analyze the stress–strain changes before slope cutting and when the slope cutting height was 7 m (Figure 7). Before cutting the slope, the horizontal stress value at the top of the slope was about 9.6 kPa, and the horizontal stress value at the toe of the slope was about 19.2 kPa. The maximum shear strain appeared inside the slope, and the slope was in a stable state. When the cutting height was 7 m, the horizontal stress at the top of the slope changed slightly, while the stress value at the toe went up to 86.2 kPa; thus, stress concentrated at the toe. The maximum strain value appeared at the bottom of the cut slope, and the most unstable area was within 1 m near the bottom of the cut slope. With the increase in the cutting height, the potential shear fracture surface moved to the inner area of the slope, and the thickness and volume of the sliding body increased.

Figure 8 presents the stress–strain distribution of the rock–soil mixed slope with a natural slope gradient of 20°. Before cutting the slope, the horizontal stress at the top of the slope was about 26.2 kPa, and the horizontal stress at the toe was about 10.2 kPa. The horizontal stress at the top of the slope was greater than that at the toe. The maximum shear strain occurred near the top of the slope, which means that the top of the slope is unstable. When the slope cutting height was 7 m, the horizontal stress at the top of the slope changed slightly by about 33.2 kPa, while the stress value at the toe increased to 66.4 kPa; thus, stress concentrated at the toe. The maximum strain value appeared at the soil–rock intersection on the cut slope surface, and the most unstable area ranged from 2 to 10 m in the direction of the soil–rock contacting surface. During the process of cutting, the rock–soil mixed slope experienced two events: soil and rock excavation. The redistribution of stress and strain in the soil excavation stage was the same as that in the soil slope, and after excavating the bedrock, the rock–soil contact area became highly unstable.

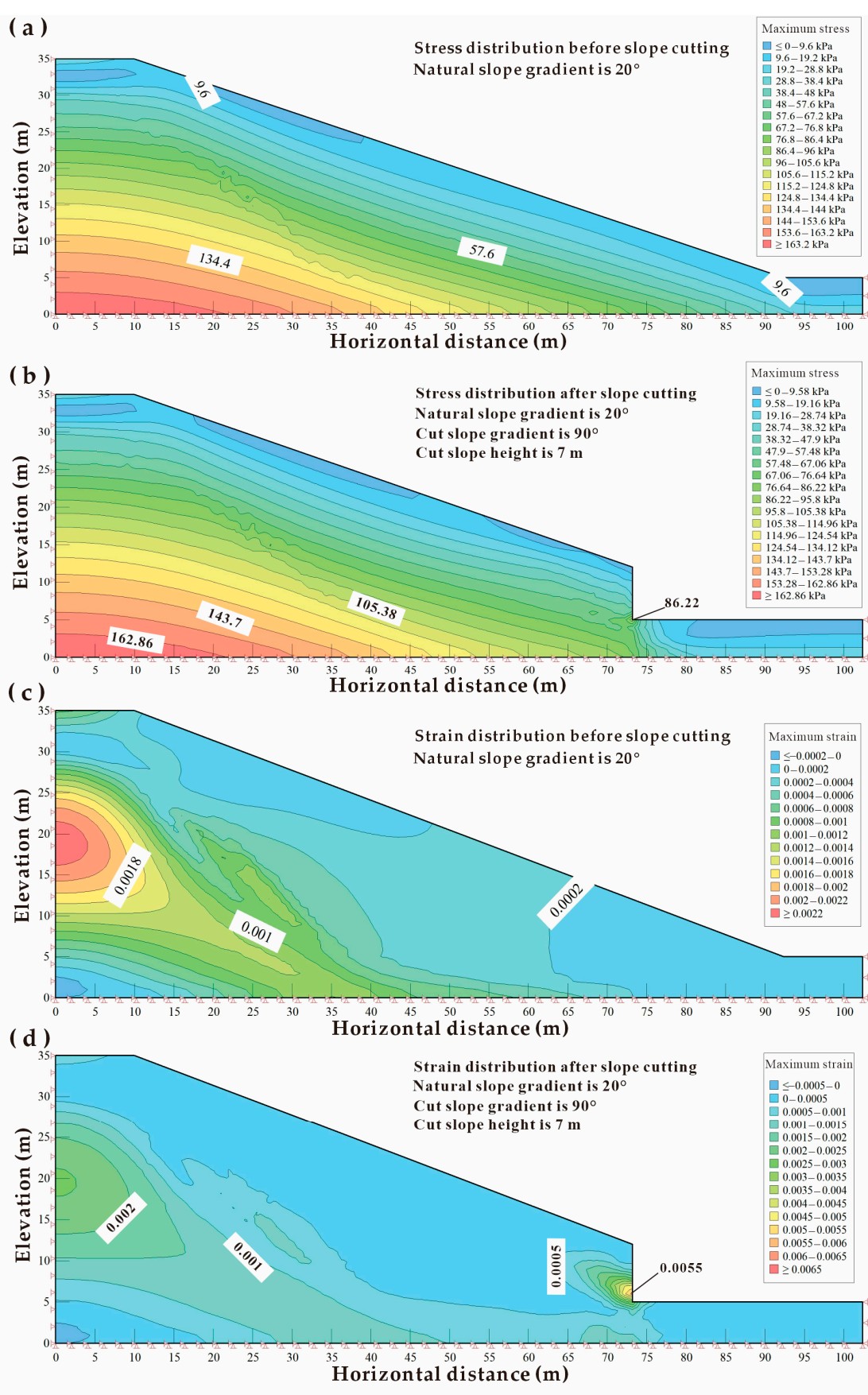

**Figure 7.** Stress and strain distribution of soil slope caused by slope cutting (natural slope gradient is 20). (**a**) Stress distribution before slope cutting; (**b**) stress distribution after slope cutting; (**c**) strain distribution before slope cutting; (**d**) strain distribution after slope cutting.

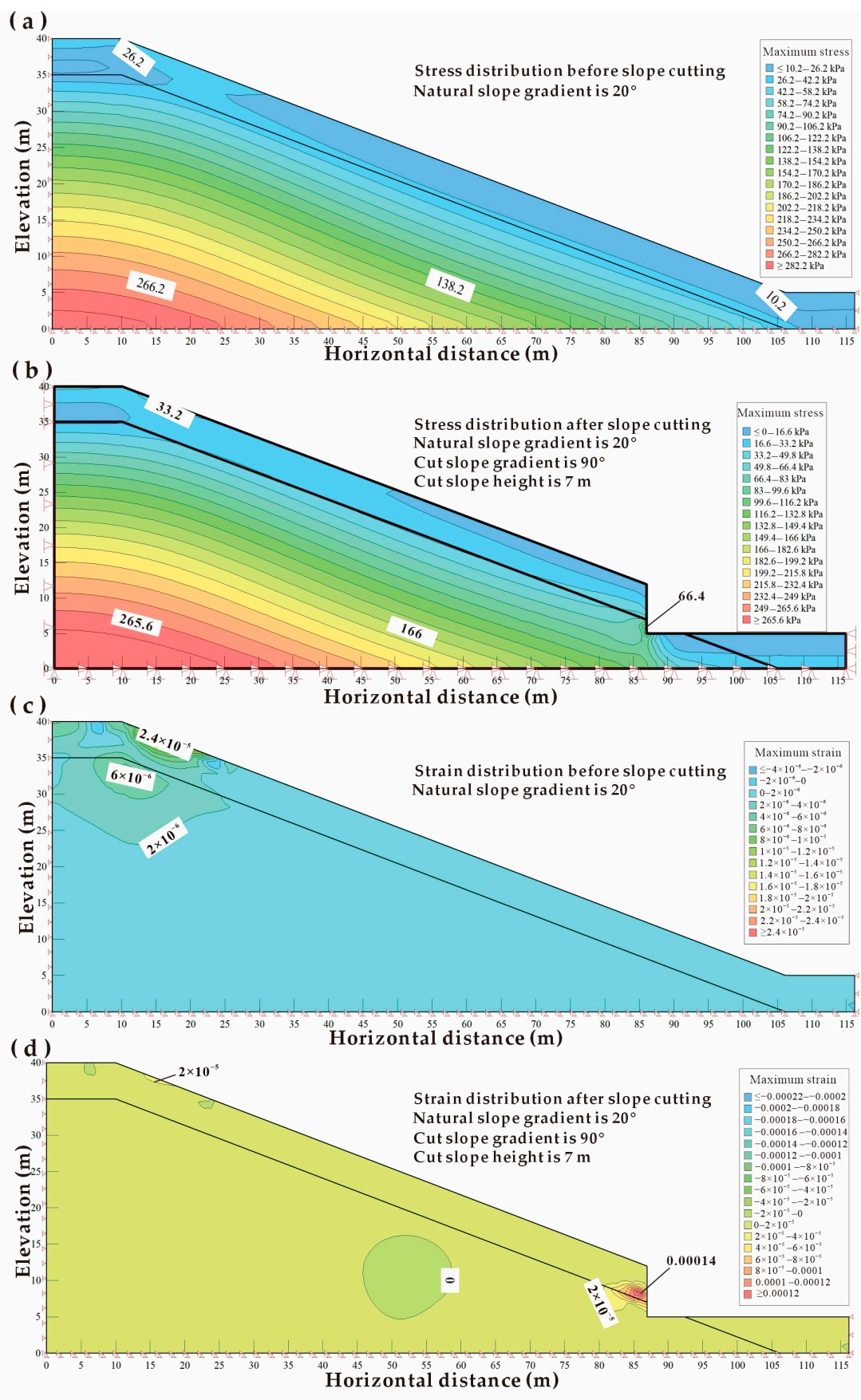

**Figure 8.** Stress and strain distribution of rock–soil mixed slope caused by slope cutting (natural slope gradient is 20). (**a**) Stress distribution before slope cutting; (**b**) stress distribution after slope cutting; (**c**) strain distribution before slope cutting; (**d**) strain distribution after slope cutting.

## 5. Discussion

### 5.1. Disaster-Causing Cut Slope Failure Mode

The main type of disaster caused by rural housing construction slope cutting in this study area is small-volume, shallow landslides. The main failure modes include: (1) the collapse of soil and rock at the top of the cut slope; (2) the creeping of residual soil on the surface of the slope; and (3) the sliding of the Quaternary overburden down the rock–soil interface.

Figure 9a presents the disaster-causing failure mode of the soil slope. The shape of the natural slope is changed by slope cutting, and it loses the support of the toe. Stress concentrates at the bottom of the cut slope face (Figure 7b), and the top of the cut slope face creeps gradually, even presenting small-scale damage or a collapse. Slope cutting destroys the vegetation on the surface of the original slope, and the soil hidden inside it is exposed. Under the continuous influence of weathering, the physical–mechanical properties of the soil continue to worsen, the cut slope gradually becomes unsteady, and lastly, a shallow landslide will happen. Over time, the upper part of the slope fails bit by bit, and the landslide materials accumulate on the new slope surface or at its foot. The gradient of the new slope is smaller than that of original, and the slope becomes more stable.

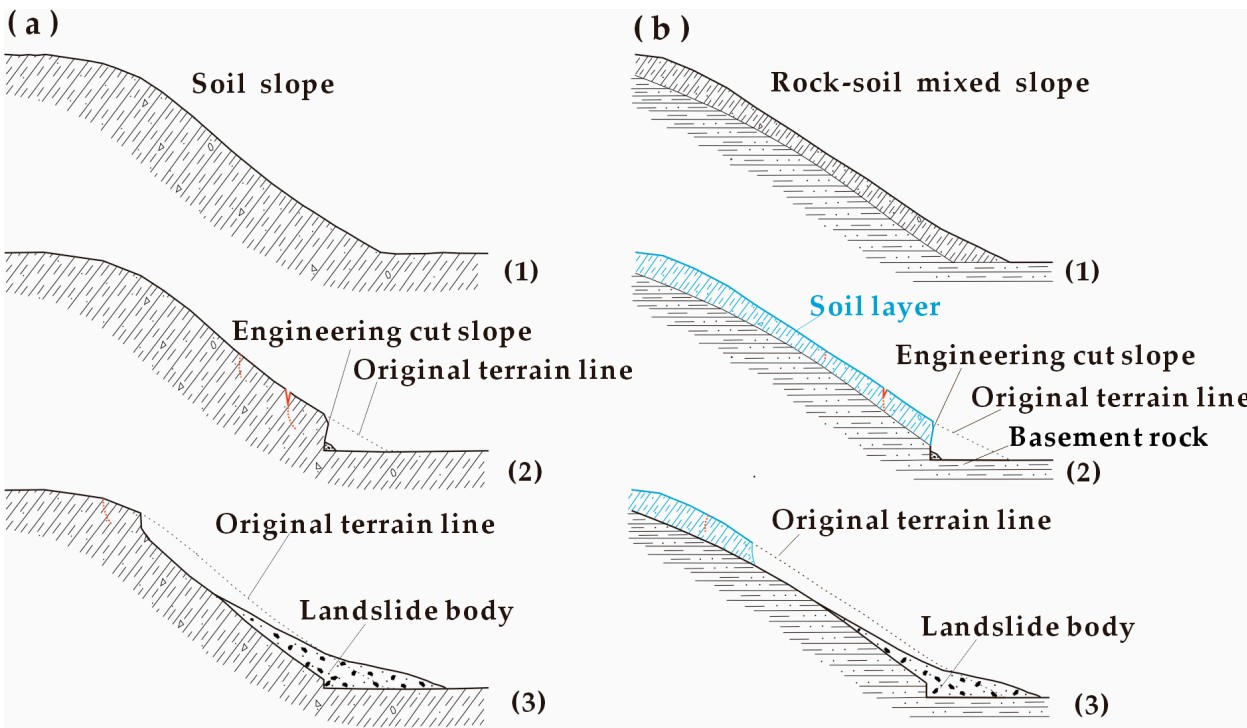

**Figure 9.** Disaster-causing cut slope failure modes. (**a**) Failure mode of soil slope; (**b**) failure mode of rock–soil mixed slope. (1) The shape of the natural slope before slope cutting; (2) slope failure after slope cutting; (3) final probable equilibrium state after slope cutting.

Figure 9b presents the disaster-causing failure mode of the rock–soil slope. The overburden is stable before slope cutting (Figure 8a). After cutting the slope, the overlying soil layer loses the support of that at the foot of the slope and gradually changes as follows: (1) deformation and small-scale collapses happens at the top of cut slope; (2) the deformation and displacement of the front edge of the overlying soil layer gradually increase, and tension cracks that appear at the middle–upper position of the overlying soil layer expand inward; (3) due to various internal and external factors, such as gravity and rainfall, the cracks extend deeply and reach the rock–soil interface, and the overlying soil layer breaks into small pieces and slides down; (4) the upper region of the slope fails bit by bit, and the soil layer blocks slide down piece by piece till the slope becomes stable.

### 5.2. Influence of Slope Cutting Height on Slope Stability

SPSS software (2018 R2) was utilized to fit the slope cutting height and stability coefficient in Table 3, and the fitting function curves are shown in Figure 10. In the figure, the ordinate is the stability coefficient, and the abscissa is the slope cutting height. On the whole, the cutting height and stability coefficient of both the soil and rock–soil mixed slopes are negatively correlated. For the soil slope, with the increase in the cutting height, the stability coefficient of 10° reduces quickly, and then slowly, while those of 20°, 30°, and 40° reduce slowly, and then quickly; the rate of the 30 natural slope gradient is the gentlest. For the rock–soil mixed slope, with an increase in the cutting height, the stability coefficients of the 10° and 20° slopes decrease relatively quickly, while those of the 30° and 40° slopes decrease relatively slowly.

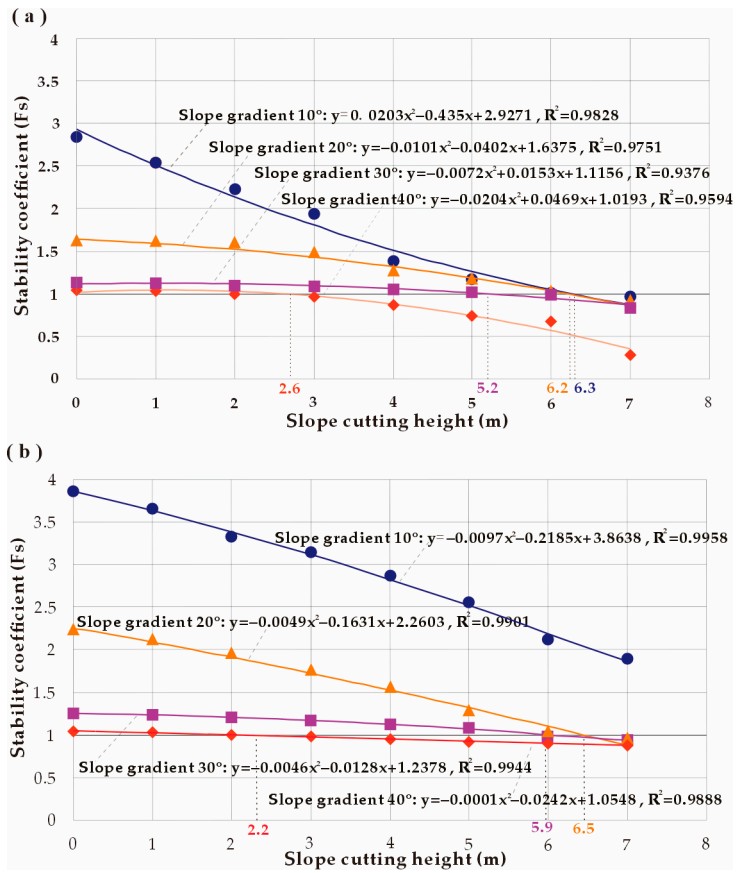

**Figure 10.** Fitting curves of slope cutting height and stability coefficient. (**a**) Fitting curves of soil slope; (**b**) fitting curves of rock–soil mixed slope.

### 5.3. Critical and Recommended Values of Slope Cutting Height

It is stipulated in the landslide control engineering survey that the stability coefficient of the slope under natural conditions is Fs $\geq$ 1.25. In this paper, Fs = 1.0 and Fs = 1.25 were determined by the critical and recommended values, respectively, of the slope cutting height of two typical slopes, as shown in Table 4. When the natural slope gradient is 30 or 40, there is no recommended value for the slope cutting height; thus, the slope is very unstable, no matter how low the cutting height is, and it is not safe to build houses by cutting slope in this situation.

**Table 4.** Critical value and recommend value of cut slope height.

| Slope Type | Gradient of Natural Slope | Fitting Function of Cut Slope Height and Stability Coefficient | Critical Value of Cut Slope Height (m) | Recommended Value of Cut Slope Height (m) |
|---|---|---|---|---|
| Soil slope | 10° | $y = 0.0203x^2 - 0.435x + 2.9271$ | 6.3 | 5.0 |
| | 20° | $y = -0.0101x^2 - 0.0402x + 1.6375$ | 6.2 | 4.5 |
| | 30° | $y = -0.0072x^2 + 0.0153x + 1.1156$ | 5.2 | no solution |
| | 40° | $y = -0.0204x^2 + 0.0469x + 1.0193$ | 2.6 | no solution |
| Rock–soil mixed slope | 10° | $y = -0.0097x^2 - 0.2185x + 3.8638$ | 9.3 | 8.6 |
| | 20° | $y = -0.0049x^2 - 0.1631x + 2.2603$ | 6.5 | 5.3 |
| | 30° | $y = -0.0046x^2 - 0.0128x + 1.2378$ | 5.9 | no solution |
| | 40° | $y = -0.0001x^2 - 0.0242x + 1.0548$ | 2.2 | no solution |

In order to verify the consistency between the fitting function of SPSS software and the Geo Studio simulation results, the critical values of the determined slope cutting height were used to reconstruct the numerical model for verification. The comparative analysis results are shown in Table 5. For the soil slope, the difference values of stability coefficients determined by the fitting function and re-simulated by Geo Studio software, respectively, were within 5%; thus, the fitting functions were in good agreement with the numerical simulation model. For the rock–soil mixed slope, when the gradients of the natural slope were 20°, 30°, and 40°, the different values determined by the fitting function and re-simulated by the Geo Studio software, respectively, were all within 7%. The reliability of the critical cut slope height was good; however, there was a big difference in the stability coefficients determined by the fitting function and re-simulated by the Geo Studio software when the natural slope gradient was 10°. Thus, this method was not suitable for determining the critical and recommended values of the slope cutting height in this situation.

**Table 5.** Comparison of stability coefficient determined by fitting function and re-simulated by Geo Studio software.

| Slope Type | Gradient of Natural Slope | Critical Value of Cut Slope Height Determined by Fitting Function (m) | Stability Coefficient Determined by Fitting Function | Stability Coefficient Re-Simulated by Geo Studio Software | Comparison of Two Kinds of Stability Coefficient |
|---|---|---|---|---|---|
| Soil slope | 10° | 6.3 | 1.0 | 1.025 | 2.44% |
| | 20° | 6.2 | 1.0 | 1.035 | 3.38% |
| | 30° | 5.2 | 1.0 | 1.023 | 2.25% |
| | 40° | 2.7 | 1.0 | 0.991 | −0.90% |
| Rock–soil mixed slope | 10° | 9.3 | 1.0 | 2.152 | 53.53% |
| | 20° | 6.5 | 1.0 | 1.075 | 6.98% |
| | 30° | 6.0 | 1.0 | 1.021 | 2.06% |
| | 40° | 2.2 | 1.0 | 0.993 | −0.70% |

### 5.4. Engineering Implications

According to the previous research in this paper, the following comments are provided to sustainably develop and manage the land resources and prevent geological hazards of cut slopes caused by rural housing construction in red bed hilly areas.

1. Before cutting the slope, the geological data should be collected, and a detailed engineering survey should be carried out to obtain topographic data, a geological investigation on internal and external factors affecting cut slope stability must be conducted carefully, and detailed engineering plans and sections should be drawn out; a safe and feasible slope cutting scheme should be formulated. Because the slope cutting project has certain dangers, it is necessary to hire a professional construction team to cut slopes.

2. As shown in Table 4. For soil slope, the critical cut height values were 6.3 m, 6.2 m, 5.2 m, and 2.6 m at slope of 10°, 20°, 30° and 40°, respectively. For the rock–soil mixed slopes, the critical cut height values were 6.5 m, 5.9 m, and 2.2 m at a slope of 20°, 30°

and 40°, respectively. If the actual cut height value is exceeded, the slope may become unstable. When the natural slope gradient is 30 or 40, there is no recommended value for the slope cutting height. In addition, the critical and recommended values are suitable when the slope gradient and lithology composition are simple. And if the topography or lithology is complex, it is necessary to construct a detailed numerical model and simulate actual cut slope stability.

3.  After cutting the slope, the protective measures in the slope cutting area should be paid attention to. The protective measures should be based on the principle of prevention and combine environmental protection and engineering treatment. Protective measures should be based on the stability requirements of the cut slope and refer to the existing local engineering experience. The construction scheme should be economical and feasible, with strong operability. Protective measures should pay attention to the drainage and treatment of surface water and groundwater. The cut area should be strengthened by using a dry blocks, concrete retaining wall, mortar blocks, or other methods.

## 6. Conclusions

The main factors affecting the stability of the cut slopes were summarized through a field investigation and the analysis of slope cutting sites caused by rural housing construction in the red bed areas of Wanyuan City. The typical slope cutting sites were selected, two-dimensional numerical models were constructed, and the stress–strain distribution and stability coefficient were simulated and computed under different excavation conditions. The disaster-causing cut slope failure modes were speculated, and the critical and recommended values of slope cutting heights under different excavation conditions were determined. The main conclusions are drawn as follows.

1.  Based on the field investigation of 76 slope cutting sites caused by rural housing construction, it is found that the internal factors affecting the stability of cutting slope mainly include the slope type, lithologic composition of slope, and natural slope gradient, and the external factors include the cut slope height, cut slope gradient, and protection engineering.

2.  The stability coefficient and stress–strain distribution of the cut slope under different working conditions were obtained by numerical simulation of two typical cut slopes. It is clear that the horizontal stress and strain at the toe of the cut slope with different natural slope gradients are positively correlated with the slope cutting height. The greater the slope gradient is, the higher the height of slope cutting is, and the more the horizontal stress increases.

3.  Slope cutting greatly influenced redistribution of stress and strain in the slope body. For the soil slope, the stress at the slope toe was unloaded after slope cutting, and the maximum stress and strain values appeared in the slope toe area. The rock–soil mixed slope went through two stages: soil and rock cutting. After cutting to the bedrock surface, the maximum stress and strain values appeared at the intersection area of the soil–rock interface and slope cutting surfaces.

4.  There are three main failure modes of shallow landslide caused by slope cutting: (1) the collapse of soil and rock at the top of the cut slope; (2) the creeping of residual soil on the surface of the slope; and (3) the sliding of the Quaternary overburden down the rock–soil interface. Slope cutting activities accelerate the evolution process of slope landform.

5.  In this paper, the fitting functions of the cutting height and stability coefficient were used to determine the critical and recommended values of the slope cutting height. For the soil slope, the critical cut height values were 6.3 m, 6.2 m, 5.2 m, and 2.6 m at a slope of 10°, 20°, 30° and 40°, respectively. For the rock–soil mixed slopes, the critical cut height values were 6.5 m, 5.9 m, and 2.2 m at a slope of 20°, 30°, and 40°, respectively. Generally speaking, the fitted function relationship could be used to predict the stability of a slope after slope cutting.

**Author Contributions:** Conceptualization, H.H.; methodology, H.H. and X.D.; validation, S.D. and H.G.; investigation, G.C. and Y.Y.; writing—original draft preparation, H.H.; writing—review and editing, H.H. All authors have read and agreed to the published version of the manuscript.

**Funding:** This work was not supported by any funds.

**Informed Consent Statement:** Not applicable.

**Data Availability Statement:** Restrictions apply to the availability of these data. Partial data were obtained from local government department and are available from the authors with the permission of the local government.

**Acknowledgments:** The authors would like to thank all the participants in the research activities.

**Conflicts of Interest:** Authors Hailin He, Simin Du, Hua Guo, Yue Yan and Guohui Chen are affiliated with The 1st Geological Brigade of Sichuan. The remaining author declares that the research was conducted in the absence of any commercial or financial relationships that could be construed as a potential conflict of interest.

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
