# Peer review of "Study on the Stability of Cut Slopes Caused by Rural Housing Construction in Red Bed Areas: A Case Study of Wanyuan City, China"

_sustainability, doi:10.3390/su16031344_

Round 1

Reviewer 1 Report

Comments and Suggestions for Authors

In this study, the authors reported a study on the stability of cut slope caused by rural housing construciton in red layer areas: a case study of Wanyuan city, China. However, the manuscript contains a lot of shortcomings. Detailed comments are as follows:

(1) The abstract is not detailed enough. The abstract should show the theoretical novelty and     practical significance. And the summary of the conclusions is incomplete. 

(2)The conclusions are not comprehensive enough. It is necessary to more clearly show the novelty of the article and the advantages of the proposed approach. The conclusions presented are only a description of the test results.

(3)What are the innovations of this research, what problems can it solve, and what engineering significance does it have? Please add in the Discussion section.

(4)Important, Results section is missing. What results did the author get?

(5)How to determine model parameters? The geotechnical parameters which were obtained via field engineering drilling and indoor geotechnical tests were used in numerical simulation. Can indoor test results be directly applied to numerical simulations? Why?

Author Response

Dear Reviewer,

Thank you for your helpful and careful comments concerning our manuscript entitled “Study on the stability of cut slopes caused by rural housing construction in red bed areas: a case study of Wanyuan city, China” (ID: sustainability-2789904). The quality of our manuscript will be greatly improved under your guidance. Based on your suggestions, we have carefully revised the paper. Thank you very much!

After studying the comments and suggestions carefully, we have made revisions which we hope meet with approval. The responds to the comments and main corrections are as following.

Comment 1: The abstract is not detailed enough. The abstract should show the theoretical novelty and practical significance. And the summary of the conclusions is incomplete.

Reply 1: Thank you very much for your important and helpful comment. According to this comment, we have modified the abstract (Line # 22-28) and conclusions (Line # 431-435, Line # 440-444).

Comment 2: The conclusions are not comprehensive enough. It is necessary to more clearly show the novelty of the article and the advantages of the proposed approach. The conclusions presented are only a description of the test results.

Reply 2:

Thank you very much for this comment. According to this comment, we have modified the conclusions (Line # 431-435, Line # 440-444).

Comment 3: What are the innovations of this research, what problems can it solve, and what engineering significance does it have? Please add in the Discussion section.

Reply 3: Thank you very much for this comment. We have added “5.4 Engineering implications” in “5 Discussion” (Line # 364-392) to highlight the significance of this paper in engineering.

Comment 4: Important, Results section is missing. What results did the author get?

Reply 4: We are very grateful for your comments. We have added “4 Results” (Line # 247-294).

Comment 5: How to determine model parameters? The geotechnical parameters which were obtained via field engineering drilling and indoor geotechnical tests were used in numerical simulation. Can indoor test results be directly applied to numerical simulations? Why?

Reply 5: Thank you very much for your question. We have added “3.2 geological parameter values” in “3.Materials and Methods”(Line # 189-202) to introduce the geotechnical parameters presented in table 1. We tried to find the critical and recommend cut height value that suit red bed slope in Wanyuan city, so the average value were used in this paper.

Reviewer 2 Report

Comments and Suggestions for Authors

The Authors summarizes the main factors affecting the stability of the cut slopes in the red bed areas of Wanyuan City, through field investigation and the analysis of slope cutting sites. Furthermore, two-dimensional numerical models were constructed for selected typical slope cutting sites, and the stress–strain distribution and stability coefficient were simulated and computed under different excavation conditions. Finally, the Authors speculate on the disaster-causing cut slope failure modes, and the critical and suggest recommended values of slope cutting heights under different excavation conditions.

The topic is virtually suitable for publication in "Sustainability".

The paper is quite well-written and the authors express clearly enough the purposes of the study and its application. The topic has been addressed in an appropriate manner and in the context of previous literature, describing the hypotheses and results in terms of the current state of the field. 

The topic addressed in the paper, however, is not particularly new, nor are the methods used in the simulations. There is a large scientific literature on the subject, as demonstrated by the bibliography. In other words, in my honest opinion this paper - more than a scientific article - appears as a technical note. The Authors themselves declare, in the last paragraph of the abstract that: "The findings can be used to prevent and manage the hazards caused by slope cutting in this study area", thus highlighting the extremely local value of the results obtained. For this reason, I am not sure that its application in other geological contexts will have a great impact, without detracting from its regional value.

For these reasons, originality and contribution to scholarship, as well as overall merit, are to be considered low. 

One of my main concerns is the geotechnical parameters used in the numerical simulations. The Authors declare that they were obtained via "field engineering drilling and indoor geotechnical test". 

I assume that the data in Table 1 are representative averages of the two main lithotypes, obtained from a number of samples. In the text, however, no information is given on their number or areal distribution.

Furthermore, I have some doubts whether it is correct to use only one average value ​​for each of the two main lithotypes, because I imagine that over an area of several thousand square kilometres such as the studied one, ther can be a considerable variability. 

I found figure very powerful and illuminating, but the terminology used in the captions should be aligned with the texts on the images: it should be noted that, while the word 'slop' is always used in the figures, in the captions (and more generally throughout the text), only the word 'slope' is used, which in my opinion is preferable. 

Row 321-325: The authors make no assumptions about the reasons why "there is a large difference in the stability coefficients determined by the fitting function and simulated by the Geo Studio software when the natural gradient of the slope was 10°". 

They merely state that "this method was not suitable for determining the critical and recommended values of the slope shear height in this situation": but why should it be suitable in the other cases?

References No. 9, 24 and 25: please check the correct formatting of the references and their citations in the text.

Data availability statement: the case of "data available on request" refers to restrictions (e.g. privacy or ethical), which should be clearly highlighted. Please insert reason why the data are not publicly available.

Comments on the Quality of English Language

Although I am not an english-native speaker, I found the the English language appropriate, understandable and good enough for the most of the parts.

Author Response

For research article

Response to Reviewer 7 Comments

1. Summary

Thank you for your pertinent comments on our manuscript entitled “Study on the stability of cut slopes caused by rural housing construction in red bed areas: a case study of Wanyuan city, China” (ID: sustainability-2789904). The quality of our manuscript will be greatly improved under your guidance. Based on your suggestions, we have carefully revised the paper. Please find the detailed responses below and the corresponding revisions/corrections highlighted/in track changes in the re-submitted files.

2. Questions for General Evaluation

Reviewer’s Evaluation

Response and Revisions

Is the content succinctly described and contextualized with respect to previous and present theoretical background and empirical research (if applicable) on the topic?

Yes

Are all the cited references relevant to the research?

Yes

Are the research design, questions, hypotheses and methods clearly stated?

Yes

Are the arguments and discussion of findings coherent, balanced and compelling?

Can be improved

For empirical research, are the results clearly presented?

Can be improved

Is the article adequately referenced?

Yes

Are the conclusions thoroughly supported by the results presented in the article or referenced in secondary literature?

Can be improved

3. Point-by-point response to Comments and Suggestions for Authors

Comments 1: The Authors summarizes the main factors affecting the stability of the cut slopes in the red bed areas of Wanyuan City, through field investigation and the analysis of slope cutting sites. Furthermore, two-dimensional numerical models were constructed for selected typical slope cutting sites, and the stress–strain distribution and stability coefficient were simulated and computed under different excavation conditions. Finally, the Authors speculate on the disaster-causing cut slope failure modes, and the critical and suggest recommended values of slope cutting heights under different excavation conditions.The topic is virtually suitable for publication in "Sustainability". The paper is quite well-written and the authors express clearly enough the purposes of the study and its application. The topic has been addressed in an appropriate manner and in the context of previous literature, describing the hypotheses and results in terms of the current state of the field.

Response 1: Thank you very much for your pertinent comments on some contents of our manuscript.

Comments 2: The topic addressed in the paper, however, is not particularly new, nor are the methods used in the simulations. There is a large scientific literature on the subject, as demonstrated by the bibliography. In other words, in my honest opinion this paper - more than a scientific article - appears as a technical note. The Authors themselves declare, in the last paragraph of the abstract that: "The findings can be used to prevent and manage the hazards caused by slope cutting in this study area", thus highlighting the extremely local value of the results obtained. For this reason, I am not sure that its application in other geological contexts will have a great impact, without detracting from its regional value .For these reasons, originality and contribution to scholarship, as well as overall merit, are to be considered low.

Response 2: Thank you for this sincerely comment. We have added “5.4 Engineering implications” in “5 Discussion” (Line # 341-369) to highlight the significance of this paper in engineering.

In the past, most of the researches were focused on red-bed slopes/landslides with large scale or threatening assets, and less attention was paid to manual slope cutting with scattered distribution, small scale and large number. First of all, according to statistics, there are more than 5,000 slope cutting sites caused by rural rural housing construction in Sichuan Province, China. These slope cutting sites are scattered in space, small in scale, with great differences in geological conditions, and lots of cut slopes are not strengthen by supporting projects, which directly threaten safety of houses. Therefore, this paper is of significance in improving the geological disaster management ability of local governments. Secondly, in the process of solving specific problems, we should not stick to theory and technology, we should start with the knowledge and skills we have mastered, and gradually improve the methods. We believed that as long as the goals are clear, more research methods and techniques will be adopted, and the problems will be solved gradually.

Comments 3:One of my main concerns is the geotechnical parameters used in the numerical simulations. The Authors declare that they were obtained via "field engineering drilling and indoor geotechnical test".I assume that the data in Table 1 are representative averages of the two main lithotypes, obtained from a number of samples. In the text, however, no information is given on their number or areal distribution.Furthermore, I have some doubts whether it is correct to use only one average value for each of the two main lithotypes, because I imagine that over an area of several thousand square kilometres such as the studied one, ther can be a considerable variability.

Response 3: Thank you very much for this question. We have added “3.2 geological parameter values” in “3.Materials and Methods”(Line # 189-202) to introduce the geotechnical parameters presented in table 1. In order to ensure the applicability of the numerical simulation analysis results in the study area, we collected the geotechnical parameter data in the study area as much as possible. Some of the data was obtained from field investigation and drilling samples, and the other part was collected from local government departments. What kind of geotechnical parameter value should be more appropriate to build the numerical model is a problem worthy of in-depth discussion. Is the maximum, minimum or average value more in conformity with the actual situation in the study area? Next, on the basis of investigating a large number of engineering cases, combined with numerical simulation analysis methods, we will discuss and study this problems in depth. In this paper, we use the average value of geotechnical parameters in study area to establish the numerical models.

Comments 4: I found figure very powerful and illuminating, but the terminology used in the captions should be aligned with the texts on the images: it should be noted that, while the word 'slop' is always used in the figures, in the captions (and more generally throughout the text), only the word 'slope' is used, which in my opinion is preferable.

Response 4: Thank you very much for your kind reminder. We have modified the mistakes in all the Figures ,especially in Figure 1, 2, 5, 6, 7, 8, 9.

Comments 5: Row 321-325: The authors make no assumptions about the reasons why "there is a large difference in the stability coefficients determined by the fitting function and simulated by the Geo Studio software when the natural gradient of the slope was 10°".They merely state that "this method was not suitable for determining the critical and recommended values of the slope shear height in this situation": but why should it be suitable in the other cases?

Response 5: Thank you very much for this question. The question is very instructive. In view of this problem, we have conducted analysis and discuss deeply.

Firs of all,in the process of establishing the numerical model of rock-soil mixed slope, the vertical height of slope is set, and the slope gradient is determined according to the horizontal distance from the top of slope to the toe of slope. For the rock-soil mixed slope with a gradient of 10 degrees, the vertical height is set at 40 meters and the horizontal distance is about 227 meters. From the perspective of topography, this is a long and gentle slope. After the rock and soil mass at the toe of the slope is excavated, the covering soil layer at the middle and rear position of the slope will have a drag effect on the soil layer at the top of the cut site.

Secondly, the overburden in red bed area is mainly residual slope soil, and the contact surface between soil layer and rock is the most unfavorable structural plane of rock-soil mixed slope, and its gradient is basically the same as that of topography. When the slope gradient is 10°, the contact surface between rock and soil is also close to 10°, which is far less than the internal friction angleφ value of overlying soil. Theoretically, the cut slope will in a stable state after slope cutting.

Comments 6: References No. 9, 24 and 25: please check the correct formatting of the references and their citations in the text.

Response 6: Thank you for this comment. We have checked and revised references No. 9, 24 (Line # 477-478, Line # 508-509). References No. 25 is a Chinese document, and this document does not provide an English index, so we decided to remove this document from the reference.

Comments 7:Data availability statement: the case of "data available on request" refers to restrictions (e.g. privacy or ethical), which should be clearly highlighted. Please insert reason why the data are not publicly available.

Response 7: Thank you for this suggestion. We have modified the data availability statement (Line # 452-454). Partial data were obtained from local government department and are available from the authors with the permission of local government department.

Reviewer 3 Report

Comments and Suggestions for Authors

1. The methodology combines field investigations of 76 slope cutting sites, detailed analysis of 31 disaster-causing sites, and numerical modeling using Geo Studio software​​. However, the selection criteria for the 31 sites analyzed in detail are not clearly stated, which could introduce selection bias.

2. The paper is generally well-organized and clear, but some sections could be more coherently presented.  For instance, the methodology section could be more detailed in explaining the selection criteria for the sites and the modeling assumptions.

3. Limited generalizability due to the specific geological context and potential selection bias in site analysis.  The methodology could be described more transparently.

4. Acceptance after minor revisions, particularly focused on clarifying methodology details and enhancing the general discussion of findings within the broader context of slope stability research.

Author Response

Dear Reviewer,

Thank you for your pertinent comments on our manuscript entitled “Study on the stability of cut slopes caused by rural housing construction in red bed areas: a case study of Wanyuan city, China” (ID: sustainability-2789904). The quality of our manuscript will be greatly improved under your guidance. Based on your suggestions, we have carefully revised the paper. Thank you very much!

After studying the comments and suggestions carefully, we have made revisions which we hope meet with approval. The responds to the comments and main corrections are as following.

Comment 1: The methodology combines field investigations of 76 slope cutting sites, detailed analysis of 31 disaster-causing sites, and numerical modeling using Geo Studio software. However, the selection criteria for the 31 sites analyzed in detail are not clearly stated, which could introduce selection bias.

Reply 1: Thank you very much for this comments. We are sorry that we didn't clearly explain the principle of selecting slope cutting sites in this article.

First of all, we conducted a full coverage geological survey of all 76 slope cutting sites caused by rural housing construction in red beds of Wanyuan city, collected the data of slope cutting gradient, slope cutting height and the effectiveness of strengthened project, and made a safety evaluation of each slope cutting site at the same time.

Secondly, after filed investigation, we found that 31 of the 76 slope cutting sits had disaster records, so we carried out in-depth geological work on these 31 sites, including engineering surveying and mapping, engineering drilling and trench exploration, as well as indoor test.

Thirdly, in order to ensure the applicability of the simulation results in the study area, in this paper, we decided to use the average value of geological parameters for numerical simulation analysis. These parameters are partly from this sample test results and partly collected from local government departments.We have added “3.2 geological parameter values” in “3.Materials and Methods”(Line # 189-202) to introduce the geotechnical parameters presented in table 1.

Comment 2: The paper is generally well-organized and clear, but some sections could be more coherently presented. For instance, the methodology section could be more detailed in explaining the selection criteria for the sites and the modeling assumptions.

Reply 2: Thank you for this comment. We have added “3.1. Establishment of constitutive model” in “3. Materials and Methods” (Line # 170-188) to introduce the selection criteria for the sites and the modeling assumptions.

Comment 3: Limited generalizability due to the specific geological context and potential selection bias in site analysis. The methodology could be described more transparently.

Reply 3: Thank you very much for this comment. We have add “Figure 4. Flowchart of the methodlogy” and related text introductions ( Line # 170-176). 

Comment 4: Acceptance after minor revisions, particularly focused on clarifying methodology details and enhancing the general discussion of findings within the broader context of slope stability research.

Reply 4: Thank you very much for this comment. We have added “5.4 Engineering implications” in “5 Discussion” (Line # 364-392) to highlight the significance of this paper in engineering.

We would like to express our great appreciation to you for comments on our manuscript again.

Thank you and best regards.

Round 2

Reviewer 1 Report

Comments and Suggestions for Authors

Thank you for inviting me to evaluate the article titled 'Study on the stability of cut slopes caused by rural housing construction in red bed areas: a case study of Wanyuan city, China '. this paper took the slope cutting sites caused by rural housing construction in the red bed area of Wanyuan City as the research objects. The internal and external factors affecting the stability of cut slopes were summarized through a field investigation, and two typical slopes were selected for analysis. In this revised manuscript, authors have responded to the comments appropriately. But I still have 2 more recommendations.  

1. Provide more rock information in numerical simulation  e.g. sources in Section 3.2.

2. In the conclusion part, the content should be simplified to emphasize the advancement of this research, taking example of dividing the conclusion into several pieces and adding quantitative results.

Author Response

Dear Reviewer,

Thank you for your 2 more helpful recommendations concerning our manuscript entitled “Study on the stability of cut slopes caused by rural housing construction in red bed areas: a case study of Wanyuan city, China” (ID: sustainability-2789904). Based on your recommendations, we have carefully revised the paper.

Recommendation 1: Provide more rock information in numerical simulation e.g. sources in Section 3.2.

Reply 1:Thank you very much for this helpful comment. According to this comment, we have added rock information in Section 3.2 (Line # 199-208).

Recommendation 2: In the conclusion part, the content should be simplified to emphasize the advancement of this research, taking example of dividing the conclusion into several pieces and adding quantitative results.

Reply 2:Thank you very much for this suggestion. We have simplified the content in the conclusion part and emphasized the advancement of this research(Line # 409-436).

Reviewer 2 Report

Comments and Suggestions for Authors

I am happy that my suggestions may have contributed to the improvement of the manuscript, which in my opinion can be now accepted for publication in its current form. Congratulations!

Author Response

Dear Reviewer,

Thank you for your pertinent comments  and suggestions on our manuscript entitled “Study on the stability of cut slopes caused by rural housing construction in red bed areas: a case study of Wanyuan city, China” (ID: sustainability-2789904). The quality of our manuscript has be greatly improved under your guidance. But the open review in the review report is still “I would not like to sign my review report”. We will be very grateful if you agree to sign your review report and choose “I would like to sign my review report” option in the review system. Thank you again and best wishes!

Sincerely yours, authors .